# What is usual care for teenagers expecting their first child in England? A process evaluation using key informant mapping and participant survey as part of the Building Blocks randomised controlled trial of specialist home visiting

Michael Robling,[1] Rebecca Cannings-John,[1] Sue Channon,[1] Kerenza Hood,[1] Gwen Moody,[1] Ria Poole,[2] Julia Sanders[3]

[1]Centre for Trials Research, Cardiff University, Cardiff, UK
[2]School of Medicine, Cardiff University, Cardiff, UK
[3]School of Healthcare Sciences, Cardiff University, Cardiff, UK

**Correspondence to**
Prof Michael Robling;
roblingmr@cardiff.ac.uk

## ABSTRACT

**Objectives** We compared the US-derived Family Nurse Partnership (FNP) home visiting programme when added to usually provided health and social care for first-time teenage mothers, to usual care alone. We aimed to: establish the nature of usual care, measure service usage and assess performance bias in core usual care services.

**Design** Within trial process evaluation. Local professionals completed a survey mapping local health and social care services in seven domains. This focused on services available to young women, especially those relevant to pregnant teenagers. Descriptive data were assessed thematically to establish the range of services. Quantitative data collection with FNP supervisors enumerated service provision by site. Services identified were included in main participant trial follow-up interviews at four time points to quantify usage. Usage was described descriptively by domain. We explored predictors of health visitor visits.

**Setting** 18 partnerships of local authority and healthcare organisations in England.

**Outcomes** Descriptive framework of services. Rates of service usage reported by trial participants.

**Results** 161 separate services were identified, with multiple service models in each domain, broadly categorised as universal or specialist (eg, for teenage mothers). FNP supervisors identified 30–63 universal services per site and 22–67 specialist services. Use of core maternity care services was similar across trial arms and with only small differences in use of health visiting services. Participants accessed a wide range of services. Women who had ever been homeless, who had a higher subjectively defined social status, and poorer mental health received more visits from a health visitor.

**Conclusions** The large number of services available to teenage mothers in England may limit the incremental benefit achievable through enhanced home visiting. There was little evidence of compensatory practice, such as additional care for women in the usual care arm. Measuring usual care when trialling complex interventions is challenging and essential.

**Trial registration number** ISRCTN23019866.

### Strengths and limitations of this study

▶ The identification of sometimes multiple local stakeholders and drawing on their existing knowledge using a semistructured self-completion tool about a range of relevant services enabled us to develop a rich picture of what may be usually available care for teenagers expecting their first child.

▶ Undertaking the initial mapping exercise enabled us to develop a more informed service use inventory with greater content validity than may otherwise have been possible.

▶ The combination of professionally led key informant mapping and detailed service use recording as part of trial follow-up data collection, therefore provides a more nuanced understanding of usual care.

▶ This greater understanding of the trial's control condition enhances interpretation of trial results.

▶ However, changes over time, and within and between site differences in how services are configured, perceived and understood means that a summary statement about all locally relevant services will need to be intermittently revisited.

▶ Although we have an understanding about how services were similarly or differently accessed by intervention and control participants in the trial, the intensity and duration of individual sessions for non-Family Nurse Partnership services is not known. However, comprehensively attempting to collect such detailed data from trial participants would probably not be feasible in practice.

## INTRODUCTION

Individual, social and economic circumstances faced by teenage mothers can challenge a successful start for their children. Responding in 2006, the government in England adopted a preventative US-derived

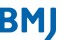

programme of nurse-led intensive home visiting, the Family Nurse Partnership (FNP). Specially trained family nurses (FNs) support first-time mothers through up to 64 home visits starting in early pregnancy and until the child reaches their second birthday. In three US trials, the programme has been evaluated with differing sociodemographic populations, justifying initial testing in a UK context.[1–3]

Following an implementation evaluation, 18 English Primary Care Trust sites participated in the Building Blocks trial (ISRCTN23019866) of the programme's effectiveness, recruiting 1645 teenagers expecting their first child.[4–8] The sites were dispersed across the UK, and covered 2 rural and 16 city areas. Women were recruited before 25 weeks gestation, lived within geographical areas served by the FNP team and spoke at least conversational English. Assessing over 60 short-term outcomes (to 24 months post partum) in domain areas of pregnancy and birth, child development and maternal life course, four primary outcomes of programme and policy interest were prioritised.

We compared FNP when added to usually provided health and social care to usual care (UC) alone. In the absence of comprehensive public healthcare in the USA, across all three previous evaluations the counterfactual was reported as obstetric office-based antenatal care, paediatric developmental screening, referral at specified time points and free transport to office-based consultations. Elevating the control condition to just more than simply no care, the augmented control condition was not further described. Given the provision of free universal health services in the UK, the ethical trial comparator was an active control condition. However, it was expected that what would be available to young families may be complex and vary by site and over time.

We aimed to map and quantify usually provided care and so clarify the trial's control condition, the service context into which FNP was introduced and allow exploration of any performance bias affecting validity of the trial comparison.

## METHODS
We first elicited and mapped usual services available locally at each of the 18 trial sites. Each site comprised collaborative partnerships between National Health Service organisations and local authorities. All sites had applied to the Department of Heath to be a provider of FNP including by demonstrating local clinical need and commitment to sustain local programme delivery. Sites included urban and rural settings across England and encompassed each of the 10 strategic health authorities in England. Second, we enumerated services accessed by participants in both trial arms.

### Eliciting and mapping services
A mapping tool was drafted using an Excel worksheet following discussion within the research team. This sought to identify services available for pregnant teenagers and young families across seven initial domains: midwifery, health visiting (specialist public health nurses), education, housing, social care and other services (eg, children's centres) and funding schemes specifically for young parents). This would, therefore, include services that were also universally available, such as maternity care. The tool required the site contacts to provide the title of service and a brief description. It was piloted with local coordinators at three sites who described service characteristics (eg, provider, eligibility criteria) and were debriefed by telephone interview to assess feasibility. An amended version, which incorporated completion instructions (online supplementary appendix 1), a worked example and study information, was circulated via email in the first instance to each site principal investigator (eg, the local FNP project lead and in all cases not a member of the research team) who then cascaded to local contacts across health and social care (usually managers or heads of services). By engaging with heads of services and other local professional staff (eg, housing support workers) further detail about specific services or domains were provided, including documentation on local services where available. Respondents were asked to provide details of 'routinely provided services within their local authority which may be provided to young women, but may be especially relevant to pregnant teenagers'. In parallel with obtaining information describing available services, national policies and guidelines were sourced informing on the minimum expected standard of universally available services such as maternity care and state welfare (eg, childcare vouchers). Mapping data were collected over 6 months.

Within sites and across respondents, we reviewed submitted returns to identify missing or incomplete data (ie, to identify the presence or absence of expected services/service descriptions) and followed up if necessary with local site contacts. This process was informed by documentary data provided by sites or available online. Data provided by sites were entered into NVivo V.8 and[9] analysed thematically by researchers who also involved service experts to review the developing coding framework before coming to a consensus on the final range of services available. A second round of online data collection addressing the same domains aimed to consolidate and confirm information already provided and to reduce variation that may be solely attributable to reporting bias. This comprised a structured form listing services by domain and tick boxes for respondents to indicate presence or absence. Free text ('Other') services allowed for unlisted services to be reported. Local FNP supervisors completed this form.

### Enumerating service use by trial participants
Trial participants were teenagers (aged 19 years or under at last menstrual period) expecting their first child, living in the catchment area for local FNP provision recruited before 25 weeks gestation, able to provide informed

consent and competent to converse in English.[7] Access to supportive services within each core domain was measured as part of the trial's follow-up outcome evaluation telephone interview schedule at late pregnancy, and 6, 12 and 18 months post partum.[7] These included use of childcare, primary (eg, midwifery, general practitioner (GP), health visiting) and secondary (eg, Accident & Emergency (A&E), outpatients, inpatients) healthcare attendances, sexual health (contraceptive services), formal education, Connexions (a government-funded support and advisory service for young people aged 13–19 years), support with housing and a range of additional support services. At 24 months, additional questions were asked about financial support.

Some data informed the separately reported cost-effectiveness analysis.[10] In the current analysis, we describe the pattern of core service usage (eg, health visiting, midwifery, housing) for those in both trial arms, and the level of support provided additionally via FNP (for FNP clients, the Healthy Child Programme was delivered by FNs rather than by health visitors(HVs)). Data on the latter were provided via the FNP national unit's information system. Use of services was analysed descriptively and is reported by service domain showing counts and proportions for those in the two-trial arms separately. Multivariable logistic regression was used to explore whether certain maternal characteristics collected as part of the trial's baseline assessment were associated with level of observed HV support. We created a binary variable of number of HV visits which distinguished between a standard/expected level of care (less than four visits) and enhanced care (four or more visits). Univariable associations were screened using a p<0.10 cut-off and retained in the final multivariable model. Estimates are shown as ORs and 95% CIs.

## RESULTS
### Eliciting and mapping services
Round 1 was conducted in a 6-week period from August 2009. All sites responded, with at least six individual informants contributing data per site. A varying level of detail was provided about identified services. In general spreadsheets circulated to multiple stakeholders were more comprehensively completed.

Similar services within any one domain were subsequently grouped together even if labelled differently by informants. This resulted in 161 identified services, some with similar aims. An example was that of education provided to pregnant teenagers aged under 16 years old with eight different named services. In round 2, conducted in July 2011, the 161 services were listed, categorised into 12 service domains (the original domains plus 'other services' subdivided on the basis of stage 1 responses into childcare, complex needs, Connexions, drug and alcohol, mental health, third sector, and sexual health).

The total number of services identified per site ranged from 52 to 113. These included between 26 and 53 universal services and between 22 and 86 locally available/specialist services. Services were provided by public, private and third sector organisations and collectively delivered direct care, support or guidance. Examples of Specialist and Locally available services for pregnant teenagers or younger parents are shown in table 1.

Not all universal services were reported from all sites although these would have been available (eg, universal education provision to age 16 years). In the domains of mental health, addiction and complex needs provision a small number of sites reported no additional locally available or specialist services. No sites reported specialist health visiting services for teenagers. Fourteen sites reported the employment of specialist teenage pregnancy midwives. Details from local informants describe the type and range of services available across the range of providers and sector domains. Services were numerous, complex and in some cases with fluid boundaries facilitating multidisciplinary interaction to support users. Individual services although provided with similar intent could vary by site, while administrative boundaries between services were shown to be fluid.

### Service usage during the trial
Initially, 823 women were allocated to receive FNP and 822 women to UC and following mandatory or elective withdrawal (including of consent), 808 and 810 women, respectively, completed baseline assessment.[8] The median ages (25th–75th centile) of women were 17.9 (95% CI 17.0 to 18.8) in the FNP arm and 17.9 (95% CI 16.9 to 18.8) in the UC arm. Interviews were completed with 501 women (FNP) and 466 women (UC) at 18 months. At 24-month follow-up, th number of interviews completed were 595 (FNP) and 559 (UC). The first woman was recruited to the trial on 16 June 2009 and the date of the last follow-up (24-month) assessment was 24 April 2013.

### Community health visiting, midwifery and FNP
Core publicly funded services for mothers are maternity care and health visiting. The mean number of all home visits from health visitors was similar in both study arms (UC: 5.01 (SD 5.51); FNP: 4.70 (SD 7.81)). Contact with health visitors in clinic was quite different with more reported by mothers in the UC arm (mean 6.31, SD 7.07) than in the FNP arm (0.70, SD 2.92). The number of contacts within each reporting period up to 18 months reflects a similar pattern (table 2). The mean number of community midwifery contacts during pregnancy for the 422 UC women responding in late pregnancy was 10.69 (SD: 5.34) and for the 459 in the FNP arm was 10.68 (SD: 5.25). Women allocated to FNP received an average of 9.71, 18.63 and 13.22 valid FN visits per programme phase (pregnancy, infancy, toddlerhood) with average visit duration of 79.14 min. There was a programme attrition rate by phase of 3.6%, 10.1% and 7.9%, respectively (cumulative rate of 21.1%).

**Table 1** Service mapping—examples of services* described by study sites†

| Domain | Specialist services—specifically for pregnant teenagers or younger parents | Locally available services¶—with a specialist nature and eligibility criteria, but not necessarily designed for teenage parents |
|---|---|---|
| Education | ► Schools/colleges with provision for teenage mums.<br>► Teenage pregnancy support services.<br>► Accredited courses with free child care for under 25 s. | ► Home learning programmes. |
| Housing | ► Teenage parents' scheme: training in independent living skills.<br>► Supported housing: young vulnerable women or teenage parents. | ► Outreach support service aimed at young homeless people under 18.<br>► Mother and baby hostel. |
| Health visiting | | ► Antenatal contact at home or in midwife-led antenatal clinics.<br>► Minor ailments sessions run by health visitors. |
| Midwifery | ► Teenage pregnancy midwives.<br>► Antenatal clinics run by midwives in schools. | ► Midwives based in Children's centres. |
| Social services | ► Teenage pregnancy support service. | ► Targeted youth support for vulnerable young people.<br>► Specialist therapeutic unit for young victims of sexual abuse.<br>► Family resource service; practical support to access universal services. |
| Connexions services‡ | ► Teenage pregnancy advisors help young mums to be and young families. | ► Provide information and guidance to looked after young people.<br>► Provide support and guidance for young people leaving care.<br>► Provide practical help and advice for young mums who want to go back to college. |
| Drugs, alcohol and smoking | | ► Specialist drugs and alcohol services working with police.<br>► Community-based young people's drugs and alcohol service.<br>► Smoking in pregnancy cessation service. |
| Sexual health | ► Lifestyle services working with teenage parents to prevent second pregnancy. | ► Family planning services for under 25 year olds in community settings.<br>► Sexual health services for teenagers.<br>► Condom distribution scheme in community settings. |
| Mental health services | | ► Specialist Children's and Adolescent Mental Health Services for eating disorders.<br>► Mother-and-baby units in hospitals and prisons.<br>► Specialist psychiatric unit for postnatal mental illness. |
| Complex needs services | ► Support and advocacy for (pregnant) teenagers with complex needs. | ► Child development centre for preschool children with complex needs.<br>► Sure start language therapy team.<br>► Vulnerable baby service: targeted safeguarding prevention. |
| Childcare provision | | ► Private, voluntary, independent childcare providers.<br>► Internet database on county-wide childcare provision. |

Continued

**Table 1**  Continued

| Domain | Specialist services—specifically for pregnant teenagers or younger parents | Locally available services¶—with a specialist nature and eligibility criteria, but not necessarily designed for teenage parents |
|---|---|---|
| Local/third sector projects | ► Charity-funded teen parents projects.<br>► Peer support sessions for teenage fathers to be. | ► Barnardo's Priory Family Centre.<br>► Charity-funded young parents projects.<br>► Home Start: trained volunteers visit mums for approximately 15 months. |

*Set information provided by local informants for each reported service included: name of service, narrative description, limits on availability (eg, upper limit on number of women offered service, location (eg, base), level of service provision per client (eg, frequency, duration, quantity), illustrative current caseload, delivery setting, client eligibility criteria, service provider (eg, local authority), assessment of local service variations compared with other locations.

†Data collection timing: Round 1: Data collection was requested over a 6-week period from August 2009 to coincide with early stages of trial recruitment; Round 2: The survey link was sent to local FNP supervisors for completion in July 2011.

‡A government-funded advisory and support service for young people aged 13–19 years, now discontinued.

§A tiered system of local government throughout England has responsibility for services including education, housing and social services. For example, across England there are 152 separate Local Education Authorities, each of which has responsibility for providing child education in their area. The responsibility for the provision of social services and housing will rest with either one of the 152 principal authorities or, particularly in large urban areas, devolved to 1 of 326 lower tier authorities. Until April 2013 (ie, within the time frame for the Building Blocks trial), 10 strategic health authorities existed across England, with healthcare provided through local NHS Primary Care and Hospital Trusts. Subsequent to the trial period and from 1 October 2015 the responsibility for commissioning public health services for children aged 0–5 transferred from NHS England to local authorities.

¶Locally available services would exclude universally available services, which may be provided across all sites (whether provided specifically for women of a certain age or all women). Hence, routine midwifery care (eg) would not be reported here.

NHS, National Health Service.

We explored variation in core service usage to determine whether level of observed support (3 or fewer HV home visits/4 or more HV home visits in the first 6 months post partum) was directed to participants distinguishable on the basis of baseline characteristics (table 3). Women who had ever been homeless, had a higher subjectively defined social status, and poorer mental health were associated with four or more visits, while visit frequency also varied by trial site (but was not subsequently entered into the final model) (table 3). Homelessness (OR 1.80, 95% CI 1.02 to 3.17) and subjective social status (OR 1.13, 95% CI 1.01 to 1.27) were the only two individual characteristics that remained independently associated with visit numbers.

### Other services

Participants accessed a wide range of services encompassing healthcare (table 2), housing and financial support (table 4), education, childcare and other support services including social care (table 5). A small proportion of respondents reported accessing support for housing outside of their friends and family, mostly from the local authority (table 4). The small difference in reported rates between study arms would appear to have been in part attributable to additional assistance from the FNP FN. Most participants reported being in receipt of additional publicly funded financial support. For most participants this included income support, housing benefit and council tax reductions with similar rates between study arms reported. Smaller proportions of participants reported other forms of financial assistance related to employment, education or personal health (eg, job-seekers allowance). The largest difference in reported rates between study arms was for those who received regular financial support from parents: 8.9% (FNP), 15.4% (UC).

**Table 2**  Participant reported access to health services (health visitor and contraception) by follow-up (month)

| | 6 | | 12 | | 18 | | Combined (up to 18 months) | |
|---|---|---|---|---|---|---|---|---|
| | FNP n=511 | UC n=470 | FNP n=514 | UC n=483 | FNP n=501 | UC n=466 | FNP n=501 | UC n=466 |
| **Health visitor contacts mean (SD)** | | | | | | | | |
| Home | 3.07 (6.08) | 3.35 (3.58) | 1.24 (3.67) | 1.16 (2.63) | 0.50 (2.50) | 0.93 (2.58) | 4.70 (7.81) | 5.01 (5.51) |
| Clinic | 0.51 (2.12) | 3.72 (5.04) | 0.20 (1.37) | 1.66 (2.76) | 0.06 (0.45) | 1.01 (2.51) | 0.70 (2.92) | 6.31 (7.07) |
| **Contraceptive services (%)** | | | | | | | | |
| GP surgery | 42.3 | 38.3 | 41.2 | 44.1 | 38.5 | 46.1 | | |
| Family planning clinic | 26.2 | 19.8 | 19.6 | 18.6 | 22.6 | 18.7 | | |
| Children's centre | 1.4 | 0.6 | 1.0 | 0.8 | 1.0 | 0.4 | | |
| Sexual health clinic | 6.1 | 4.5 | 4.7 | 4.3 | 7.2 | 4.5 | | |

FNP, Family Nurse Partnership; GP, general practitioner; UC, usual care.

**Table 3** Baseline predictors of number of home visits from health visitor by 6 months post partum for women in usual care arm

| | Three or fewer visits (n=155) | | Four or more visits (n=312) | | Overall n=467 | Univariate association |
|---|---|---|---|---|---|---|
| | N | Median (IQR) or % | N | Median (IQR) or % | Median (IQR) or N (%) | |
| Age in years | | 17.9 (17.1–18.7) | | 17.8 (16.9–18.9) | 17.8 (16.9–18.8) | 0.721 |
| Ethnic background | | | | | | 0.070 |
| White | 130 | 83.9 | 276 | 88.5 | 406 (86.9) | |
| Mixed | 5 | 3.2 | 18 | 5.9 | 23 (4.9) | |
| Asian | 3 | 1.9 | 4 | 1.3 | 7 (1.5) | |
| Black | 15 | 9.7 | 13 | 4.2 | 28 (6.0) | |
| Other | 2 | 1.3 | 1 | 0.3 | 3 (0.6) | |
| Relationship status | | | | | | 0.433 |
| Married | 4 | 2.6 | 3 | 1.0 | 7 (1.5) | |
| Separated | 13 | 8.4 | 34 | 10.9 | 47 (10.1) | |
| Closely involved/boyfriend | 120 | 77.4 | 244 | 78.2 | 364 (77.9) | |
| Just friends | 18 | 11.6 | 31 | 9.9 | 49 (10.5) | |
| Live with father of baby | | | | | | 0.512 |
| Yes | 42 | 27.1 | 71 | 22.8 | 113 (24.2) | |
| No | 108 | 69.7 | 212 | 67.9 | 320 (68.5) | |
| Not answered | 5 | 3.2 | 29 | 9.3 | 34 (7.3) | |
| Subjective social status: | | | | | | |
| Family | 155 | 5.8 (5.0–7.0) | 309 | 5.8 (5.0–7.0) | 5.8 (5.0–7.0) | 0.896 |
| Personal | 154 | 6.8 (5.0–8.0) | 311 | 7.1 (6.0–8.0) | 6.7 (6.0–8.0) | **0.007** |
| NEET* | 138 | | 266 | | | 0.210 |
| Yes | 45 | 32.6 | 105 | 39.5 | 150 (37.1) | |
| No | 93 | 67.4 | 161 | 60.5 | 254 (62.9) | |
| Receive any benefits | 154 | | 311 | | | 0.776 |
| Yes | 48 | 31.0 | 101 | 32.4 | 149 (31.9) | |
| No | 106 | 68.4 | 210 | 67.3 | 316 (67.7) | |
| Not answered | 1 | 0.6 | 1 | 0.3 | 2 (0.4) | |
| Ever been homeless | | | | | | **0.023** |
| Yes | 19 | 12.3 | 65 | 20.8 | 84 (18.0) | |
| No | 136 | 87.9 | 247 | 79.2 | 383 (82.0) | |
| Deprivation (IMDS)† | 154 | 40.4 (24.8–54.3) | 308 | 38.0 (24.8–51.4) | 38.8 (24.8–51.7) | 0.175 |
| Health utility | | | | | | 0.374 |
| Perfect health | 104 | 67.1 | 195 | 62.5 | 299 (64.0) | |
| Less than perfect health | 51 | 32.9 | 115 | 36.9 | 166 (35.5) | |
| Not answered | 0 | 0.0 | 2 | 0.6 | 2 (0.4) | |
| Self-rated health | | | | | | 0.227 |
| Excellent | 24 | 15.5 | 58 | 18.6 | 82 (17.6) | |
| Good | 113 | 72.9 | 200 | 64.1 | 313 (67.0) | |
| Fair | 17 | 11.0 | 48 | 15.4 | 65 (13.9) | |
| Poor | 1 | 0.6 | 6 | 1.9 | 7 (1.5) | |
| Limiting chronic illness: | | | | | | 0.144 |
| Yes | 24 | 15.5 | 66 | 21.2 | 90 (19.3) | |
| No | 131 | 84.5 | 246 | 78.8 | 377 (80.7) | |
| Self-efficacy‡ | 151 | 29.7 (27.0–32.5) | 308 | 29.9 (28.0–32.0) | 29.8 (27.0–32.0) | 0.604 |
| Adaptive functioning§ | | | | | | |

Continued

**Table 3** Continued

|  | Three or fewer visits (n=155) | | Four or more visits (n=312) | | Overall n=467 | Univariate association |
|---|---|---|---|---|---|---|
|  | N | Median (IQR) or % | N | Median (IQR) or % | Median (IQR) or N (%) |  |
| Difficulty in at least one basic skill |  |  |  |  |  | 0.674 |
| Yes | 36 | 23.2 | 78 | 25.0 | 114 (24.4) |  |
| No | 119 | 76.8 | 234 | 75.0 | 353 (75.6) |  |
| Three or fewer key life skills |  |  |  |  |  | 0.822 |
| Yes | 39 | 25.2 | 81 | 26.0 | 120 (25.7) |  |
| No | 116 | 74.8 | 229 | 73.4 | 345 (73.9) |  |
| Missing | 0 | 0.0 | 2 | 0.6 | 2 (0.4) |  |
| At least one burden†† |  |  |  |  |  | 0.080 |
| Yes | 55 | 35.5 | 87 | 27.9 | 142 (30.4) |  |
| No | 98 | 63.2 | 224 | 71.8 | 322 (69.0) |  |
| Missing | 2 | 1.3 | 1 | 0.3 | 3 (0.6) |  |
| Alcohol/drug use¶ | 147 | 1.2 (0.0–2.0) | 296 | 1.3 (0.0–2.0) | 1.3 (0.0–2.0) | 0.212 |
| Antisocial behaviour | 154 | 2.0 (1.0–3.0) | 310 | 2.3 (1.0–4.0) | 2.2 (1.0–3.0) | 0.088 |
| Social support | 155 | 85.7 (77.0–98.7) | 310 | 85.8 (79.0–98.7) | 85.8 (77.6–98.7) | 0.491 |
| Relationship quality | 130 | 28.5 (26.0–32.0) | 255 | 28.2 (26.0–32.0) | 28.3 (26.0–32.0) | 0.433 |
| Family resources | 150 | 13.5 (11.0–16.0) | 296 | 13.5 (11.0–16.0) | 13.5 (11.0–16.0) | 0.884 |
| Psychological distress/mental health | 155 | 20.3 (15.0–25.0) | 311 | 21.8 (17.0–26.0) | 21.3 (16.0–26.0) | 0.025 |
| Trial site |  |  |  |  |  | 0.003** |
| 1 | 1 | 0.6 | 10 | 3.2 | 11 (2.4) |  |
| 2 | 5 | 3.2 | 8 | 2.6 | 13 (2.8) |  |
| 3 | 14 | 9.0 | 15 | 4.8 | 29 (6.2) |  |
| 4 | 2 | 1.3 | 7 | 2.2 | 9 (1.9) |  |
| 5 | 8 | 5.2 | 10 | 3.2 | 18 (3.9) |  |
| 6 | 6 | 3.9 | 7 | 2.2 | 13 (2.8) |  |
| 7 | 7 | 4.5 | 7 | 2.2 | 14 (3.0) |  |
| 8 | 12 | 7.7 | 19 | 6.1 | 31 (6.6) |  |
| 9 | 13 | 8.4 | 26 | 8.3 | 39 (8.4) |  |
| 10 | 5 | 3.2 | 17 | 5.4 | 22 (4.7) |  |
| 11 | 7 | 4.5 | 30 | 9.6 | 37 (7.9) |  |
| 12 | 17 | 11.0 | 16 | 5.1 | 33 (7.1) |  |
| 13 | 7 | 4.5 | 35 | 11.2 | 42 (9.0) |  |
| 14 | 5 | 3.2 | 3 | 1.0 | 8 (1.7) |  |
| 15 | 11 | 7.1 | 26 | 8.3 | 37 (7.9) |  |
| 16 | 19 | 12.3 | 19 | 6.1 | 38 (8.1) |  |
| 17 | 8 | 5.2 | 30 | 9.6 | 38 (8.1) |  |
| 18 | 8 | 5.2 | 27 | 8.7 | 35 (7.5) |  |

Bold indicates variable remained significantly associated with number of visits in logistic model.
*Applicable only to those whose academic age is >16 years at baseline interview.
†Higher IMDS indicated more deprivation.
‡Higher score indicates higher level of self-efficacy.
§Higher score indicates better management of day-to-day lives and routines (for each of the three subscales).
¶CRAFFT screening test[11] for substance-related risks and problems in adolescents.
**Not modelled in regression analysis due to high number of levels.
††The three original scale items comprised having to care for someone with long-term illness or alcohol/drug problem, feeling that they had in/sufficient privacy, living with people who respondents wished were not around.
IMDS, Index of Multiple Deprivation Score; NEET, not in education, employment or training.

**Table 4** Participants (%) reporting housing and financial support by follow-up point (months)

| | 6 | | 12 | | 18 | | 24 | |
| --- | --- | --- | --- | --- | --- | --- | --- | --- |
| | FNP n=511 | UC n=470 | FNP n=514 | UC n=483 | FNP n=501 | UC n=466 | FNP n=595 | UC n=559 |
| (A) Source of housing support | | | | | | | | |
| Anyone outside of friends or family | 18.0 | 14.9 | 12.1 | 9.9 | 9.2 | 8.4 | 12.1 | 9.7 |
| Local authority housing department | 7.0 | 6.6 | 5.1 | 5.6 | 4.6 | 4.7 | 6.2 | 5.9 |
| Family nurse | 4.1 | – | 3.1 | – | 2.2 | – | 5.4 | – |
| (B) Source of financial support | | | | | | | | |
| State benefits or payments | – | | – | | – | | 86.9 | 88.4 |
| Income support | – | | – | | – | | 62.0 | 63.3 |
| Job-seekers allowance | – | | – | | – | | 8.6 | 8.9 |
| Housing benefit | – | | – | | – | | 64.2 | 68.5 |
| Council tax reduction | – | | – | | – | | 62.9 | 63.3 |
| Disability living allowance | – | | – | | – | | 2.5 | 5.4 |
| Incapacity benefit | – | | – | | – | | 0.7 | 1.6 |
| Child support agency* | – | | – | | – | | 12.8 | 11.6 |
| Regular support from parents | – | | – | | – | | 8.9 | 15.4 |
| Education grants | – | | – | | – | | 5.5 | 5.9 |

*Directly or via partner.
FNP, Family Nurse Partnership; UC, usual care.

Most women seeking contraception obtained it from their general practice, and to a lesser extent from a family planning clinic. There were some small differences between study arms by time point (eg, at 18 months 46.1% of women in the UC arm accessed contraception from their GP, while 38.5% in the FNP did) but overall use of this service was similar. The proportion of women accessing any education gradually increased across the duration of the trial. By 24 months about one-fifth of women were in school, college or training (FNP: 22.5%, UC: 18.1%). This was mostly in mainstream education, although there were a small number of women in both trial arms accessing support in more specialised units (eg, learning support unit). A similar pattern of increasing support for childcare was observed over time with approximately a quarter of women reporting some form of childcare support used at 24 months. Support was received from a variety of sources and there appeared to be a similar pattern of usage between study arms.

Various other services were accessed, the most frequent being Connexions and Children's centres. The former was used with decreasing frequency over time (consistent with the ageing profile of the sample), while the latter showed a more variable pattern of access across each time point and on occasions quite different rates of access between trial arms. At 6months, 1 in 10 mothers in both trial arms reported contact with a social worker, a rate that varied over time to 24 months at which point there was only a small difference between groups (FNP: 13.1%, UC: 9.7%).

## DISCUSSION

To understand the service context within which FNP was trialled, we mapped the range of services available. The multiplicity of services often within the same area and their varying labels often concealed similarities and differences between services. We established the usage of key services by trial participants across service domains. We particularly focused on those most directly relevant to the intervention (eg, health visiting) although included many other services. With mostly only small differences in usage between trial arms perhaps what is most important is the wide range of services being accessed. Although the previous US trials have not further reported on broader services, the contexts were likely to be very different from the English trial setting.

FNP aims to impact on a range of maternal and child outcomes. Therefore, our selection of relevant services was necessarily broad and informed by the intervention's theory of change, which includes promoting access to services. However, previously reported attempts to map services have been challenging even when restricted to a single organisation.[12] To cope with such complexity researchers have sought to distinguish between specialist

**Table 5** Participants (%) reporting access to education, childcare and other support services by follow-up point (months)

| | 6 | | 12 | | 18 | | 24 | |
|---|---|---|---|---|---|---|---|---|
| | FNP n=511 | UC n=470 | FNP n=514 | UC n=483 | FNP n=501 | UC n=466 | FNP n=595 | UC n=559 |
| **(A) Education attended†** | | | | | | | | |
| Any school, college or training | 14.5 | 16.4 | 20.4 | 19.0 | 22.4 | 20.6 | 22.5 | 18.1 |
| Mainstream school or college | 11.3 | 13.7 | 15.0 | 15.6 | 19.5 | 18.7 | 16.6 | 12.7 |
| Learning support unit | 0.6 | 0.2 | 0.6 | 0.6 | 0.2 | 0 | 0.7 | 0.7 |
| Pupil referral unit | 0 | 0.2 | 0 | 0 | 0 | 0 | 0 | 0.2 |
| Teenage mums support unit | 0.8 | 1.7 | 0.6 | 0.6 | 0.4 | 0.6 | 0.7 | 1.5 |
| **(B) Childcare accessed** | | | | | | | | |
| Any childcare | 7.0 | 7.0 | 16.1 | 13.3 | 25.5 | 21.5 | 26.9 | 24.3 |
| Crèche at school or college | 4.1 | 4.5 | 8.8 | 6.6 | 4.8 | 3.6 | 12.1 | 12.3 |
| Day nursery at children's centre | 0.8 | 0.6 | 0 | 0 | 3.6 | 2.4 | 5.5 | 4.3 |
| Child-minder | 1.8 | 1.1 | 2.1 | 1.2 | 3.2 | 2.4 | 3.2 | 3.0 |
| Other forms of childcare | 0.8 | 0.6 | 2.1 | 2.9 | 8.0 | 6.9 | 6.7 | 6.1 |
| **(C) Other support services** | | | | | | | | |
| Connexions | 31.1 | 26.8 | 23.5 | 23.2 | 16.8 | 17.0 | * | * |
| School nurse | 1.4 | 1.5 | 0.8 | 0.4 | 0 | 0.9 | 0.5 | 0.9 |
| Young people's centre | 4.9 | 7.0 | 2.7 | 3.9 | 1.8 | 1.9 | 1.8 | 1.6 |
| Family information centre | 2.0 | 2.3 | 1.2 | 1.5 | 2.2 | 3.0 | 1.3 | 1.4 |
| Children's centre | 36.6 | 36.6 | 25.8 | 35.6 | 28.3 | 30.0 | 34.6 | 26.7 |
| Child development centre | 0.6 | 0.6 | 0.4 | 1.7 | 0.8 | 1.5 | 1.0 | 2.5 |
| Crèche/day nursery | 10.8 | 10.8 | 15.4 | 14.7 | 8.4 | 6.0 | 17.6 | 16.6 |
| Toddler group | 7.8 | 7.9 | 12.5 | 11.0 | 16.2 | 15.2 | 19.2 | 21.5 |
| Leaving care service | 1.4 | 0.4 | 1.8 | 1.0 | 1.4 | 0.6 | 2.0 | 0.9 |
| Fostering service | 0.6 | 0.2 | 0.4 | 0.4 | 0 | 0.6 | 0.3 | 0.4 |
| Youth offending team | 0.8 | 0.9 | 0.2 | 0.2 | 0.4 | 0 | 0.3 | 0 |
| Social worker | 10.6 | 10.0 | 7.4 | 7.5 | 8.2 | 6.2 | 13.1 | 9.7 |
| Alcohol/drug support | 0.6 | 0 | 0.2 | 0.2 | 0 | 0.4 | 0.3 | 0.5 |

*Not collected as service reconfigured.
†Some respondents indicated they were in school, college or training but provided no further information.
FNP, Family Nurse Partnership; UC, usual care.

and generic services, including through a multistage approach as used here.[13] It has been consistently reported that information provision is time consuming for professionals (or other key informants) in such exercises, as we also found.[14] Individual informants may be unfamiliar with all relevant services even within their professional area, hence the coordinated approach to data gathering from multiple informants we used. Feedback from FNP staff in our process evaluation focus groups highlighted a similar challenge when acquiring knowledge about local services, essential for then linking up clients to relevant support.[10] Some core services such as mainstream education were not always reported and illustrates the need to clearly define the scope of the information request to informants, especially the boundaries within which they are being asked to respond. On this last point, we would also clarify that many services, however, resourced and whether universal in availability or not, may impact on the health and well-being of mother and child. We have measured for trial participants services actually used. The extent to which mothers can practically access currently unused or underused services effectively represents a key potential for future benefit if addressable barriers to access can be removed.

Our experience from this study will encourage us to further develop an approach to better understanding UC in complex service settings. Our approached spanned an elicitation phase whereby we started by plotting a map of

services and then a consolidation phase where we largely sought to confirm the contours on the map. Accordingly, we took an exploratory approach for the former and a largely confirmatory approach for the latter. How either is actually done may depend on study setting and resource. The spreadsheets worked well in that they were portable and could be transferred easily to informants for completion once we had piloted them. However, an inperson semistructured approach could have worked as well, but may have been more resource intensive. The complexity and number of services identified would have been unfeasible to include in their entirety in the trial's participant follow-up survey, but that may be important in some other studies. For example, if it was considered that sites clearly varied in provision of key services, gaining high-quality information about such site characteristics could inform more informative analysis such as multilevel modelling. Finally, we initially explored the nature of available services with professionals, and only then asked mothers about services actually used via a mostly structured list of options. An exploratory exercise with mothers may well have shed light on other potential relevant services.

In effectiveness trials existing services could respond by augmenting support to those in the control arm. Such performance bias limits generalisability especially if that support was very different from UC and approaching the level of support provided by the new intervention. Our findings do not indicate this in general and specifically for community midwifery and health visiting, the two most closely aligned universal services. However, determining only the number of contacts may mask enhanced support provided in the form of longer contacts, or contacts from specialist practitioners. Community midwives' visits were equivalent between trials arms and the difference in contacts with health visitors was attributable to clinic rather than home visits and therefore unlikely to be substantial. There was some indication that women in the UC arm with some additional objective need identified at baseline, such as experience of homelessness, received more home visits. However, providing enhanced care to clients most in need would be usual practice. Evidence that this occurred in a trial context is not in itself a threat to external validity. The large caseloads managed by health visitors emphasises the lack of opportunity to provide significant additional support to mothers allocated to UC.[10]

Our trial found fewer short-term benefits than previous US trials despite FNP being well implemented.[1–3 8] The population we studied differed from that in the USA, for example, by being fundamentally identified by maternal age and this may have contributed to some differences in impact detected. The upper age limit for women in the US trials was greater in each case than in England, and they also could have been enrolled at a later stage of gestation, for example, before delivery in Denver. In the three US trials the intervention had been provided by a total of 5 (Elmira), 12 (Memphis) and 10 (Denver) nurses in single areas with study samples of 400, 1138 and 735 women,

respectively. In our trial 131 nurses delivered the intervention across 18 local sites. The English service context would have been very different. Some additional standardised support in the form of developmental screening and referral, and free travel to appointments was provided to women in the control arm of each US trial. However, the broad and layered range of services identified in our study would not have been available. The broader adverse social context present in the first US trial and from which much longer-term evidence has been derived has limited direct comparison. For example, at the inception of the first US trial, Elmira was ranked bottom of all 380 US metropolitan statistical areas in terms of economic conditions. That is not to say that women in our trial were free of disadvantage or had services that fully met their needs. However, substantial differences across trial settings and the substantial duration between the trials are likely to have varied the potential for beneficial impact.

Service provision may change over time and any single mapping exercise will miss this real-world dynamic. We conducted telephone interviews with five FN supervisors towards the end of the trial. These explored whether there had been any key changes to local service provision. Recent major change was mostly not identified as occurring although the reduction in Connexions services was flagged up. Quantification of service use should be open to the capture of newer services. Additionally, with superficial service names not always reflecting well actual support provided it is important to look beyond service labels. Finally, high-level service descriptions do not always represent the often complex multiprofessional interactions which necessarily facilitate service delivery. This emphasises the need for adequate qualitative description and interpretation of services.

Loss to follow-up at assessment points may introduce bias into the descriptive analysis. We have previously reported on group differences in attrition apparent at 24 months follow-up, however, such differences were small.[10] A second consideration is the level of detail available for health visitor and midwifery contacts (eg, visit duration). It is reasonable to assume that given capacity and opportunity, women in the UC arm visited by health professionals would have received greater attention than other clients perceived as less in need. This is consistent with their professional role and reflective of contemporary best usual practice.[15] It is also possible that women in the FNP arm received relatively less attention than non-FNP clients if they were seen to be receiving enhanced support. Nevertheless, the total number of home and clinic visits received in both trial arms was small compared with that provided by FNP nurses. Future process evaluations should model the impact on existing services of such service innovation to both avoid unintended consequences (eg, service displacement) and maximise synergy across services.

Moore and colleagues recommend primarily qualitative methods for capturing unanticipated or complex intervention pathways, which in this instance we take to be impact on coexisting services.[16] They also emphasised

the need to capture the mechanisms using logic models including where these reflected broader context. The extent to which an intervention's impact could actually induce harm either at the individual level or within a system can further be reflected by use of a dark logic model.[17] Bonell *et al* recommend approaches to developing such a logic model, for example, by hypothesising how the agency of key stakeholders may interact with social structures to produce unintended consequences. Reflection in such model building could be informed by the use of mid-range sociological or psychological theory. This could also be combined with exploratory qualitative work with local stakeholders (eg, service managers or practitioners) well placed to observe both intended and unintended intervention impacts. This is also consistent with approaches which recognise the implementation of public health interventions occurring within complex adaptive social systems, such as May's normalisation process theory (NPT).[18] NPT identifies implementation as occurring in a dynamic, non-linear and emergent fashion. This offers a broader theoretical context within which to explore how one intervention becomes adapted to its environment and may vary, and how that social context and usual services may also become adapted too.

The effectiveness of a public health intervention can only be adequately evaluated with a sound understanding of the service context within which it operates and which may also form the trial comparator.[19] Describing and quantifying the nature of usually available services can be challenging especially when services arise from a number of sectors, may evolve over the period of study and vary across study sites. In mapping the pattern of support potentially available to participants in our trial, we have gained a critical understanding of the context within which and against which FNP should be considered. In quantifying maternal-reported service usage, we have provided key insights into how our main trial results should therefore be interpreted. While challenging, we remain convinced of the need to develop this area of research when evaluating public health interventions. Indeed, in their feedback survey respondents reported the usefulness of the exercise in gaining greater insights about local services, some sharing the generated service summaries with their teams.

**Acknowledgements** The authors gratefully acknowledge the contribution of Dr Lesley Wye, University of Bristol and Dr Marie-Jet Bekkers who contributed to some of the planning and implementation of the reported study, and to all other members of the broader Building Blocks research team who supported overall trial delivery. We are grateful to each local site informant who contributed to data collection in either of the two rounds of service mapping and especially to the three local coordinators who initially piloted our mapping tool. We are also grateful to all women who participated in the trial and for their time in completing outcome and resource assessments.

**Contributors** MR, KH and JS conceived the study and all authors contributed to the development of its protocol. MR wrote the first draft with further contributions from all authors. RP, JS and GM were involved in data collection and management. RP was responsible for developing the survey of local stakeholders, and GM was responsible for managing data collected from trial participants used in the analysis. RP, RC-J, GM and MR were involved in analysis and developing summary tables

for publication. SC and JS were responsible for the management of this package of work within the trial overall. MR was responsible for obtaining study funding. All authors contributed to data interpretation, reviewed successive drafts and approved the final version of the manuscript.

**Funding** The Centre for Trials Research, Cardiff University is funded by the Welsh Government through Health and Care Research Wales and the authors gratefully acknowledge the Centre's contribution to study implementation. This is an independent report commissioned, sponsored and funded by the Policy Research Programme in the Department of Health (reference no. 006/0060).

**Competing interests** Prior to working on the Building Blocks trial, SC facilitated two workshops on motivational interviewing for supervisors within the FNP.

**Patient consent** Not required.

**Ethics approval** The trial was approved by the Wales NHS Research Ethics Committee (09/MRE09/08) and received governance approval from all participating NHS sites.

**Provenance and peer review** Not commissioned; externally peer reviewed.

**Data sharing statement** The datasets generated and analysed during the current study are not publicly available as contributors/participants may be identifiable and are also subject to sponsor approval, but may be available from the corresponding author on reasonable request.

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
