## [Reviewer comments · BMJ Open]

ARTICLE DETAILS

TITLE (PROVISIONAL)	What is usual care for teenagers expecting their first child in England?: a process evaluation using key informant mapping and participant survey as part of the Building Blocks randomised controlled trial of specialist home visiting.
AUTHORS	Robling, Michael; Cannings John, Rebeca; Channon, Sue; Hood, Kerenza; Moody, Gwenllian; Poole, Ria; Sanders, Julia

VERSION 1 – REVIEW

REVIEWER	Janice Christie The University of Manchester Oxford Road Manchester M13 9PL
REVIEW RETURNED	10-Nov-2017

GENERAL COMMENTS	Thank you for the opportunity to review this paper. It addresses an important gap in the literature about usual postpartum care is for UK postpartum teenage and young first-time mothers. In addition, it also provides a potential method of collecting information for any trial in which there is a complex context and/or potentially wide variation in usual care. It would be helpful to clarify or add additional information about: 1) The context of postpartum care especially for teenage and young first-time mothers in the UK for the international reader. This could include a brief description/evaluation of the domains in table 1.2) Clearer conceptualisation in the paper about what usual care is and the difference between what is available versus what is used. E.g. Line 26 refers to universal health services; yet, the paper considers a wider range of services that could impact health. The mapping tool identified what was available; yet, the trial data identified what was used. It would be helpful to add a column to table 3 with the table 1/mapping tool main categories.3) Methods/sample- how many sites were mapped (this is in the abstract and it would be helpful to have a reminder around lines 35-38). Some contextual information about the sites would be great e.g. rural/urban, geographically close or dispersed (with the same Trusts) and socio-economic status. What was the age range of the trial sample and were the 'usual' services available to everyone in this age range (it appears some universal services are missing from table 1 e.g. well baby clinics and antenatal/postpartum home visits by HV and midwives)? It would be helpful to clarify if data was sought about universal services that only supported younger parents or if it also included services that were available to every parent.
--

	4) Methods/data collection- did the mapping tool require the site contacts to provide a description and title of service? How many mapping tools were initially disseminated per site and what was the role and background of the site contacts e.g. where they joint research and clinician posts? If they had a clinical role what was their role e.g. service providing health care professional or line manager or senior manager? Was data collected about how many consultations (and with whom) site contacts made to complete the form? If so was this similar or did it vary across sites? When was mapping data collected? When was the service use data collected from participants? 5) Methods/data analysis- Could 'summative content analysis' be more clearly explained? For example, does this refer to content analysis e.g. counting the occurrence of words/phrases or development of themes to group together similar types of services? 6) Methods/missing data-What were the criteria for follow up of missing information? For example was the total number of services counted and areas that looked like they had few services contacted for follow up or was this process more thematic? How much data was originally missing (e.g. how much was picked up on the 2nd round)? Was the 2nd round of mapping sent to the same people as before and did people respond to follow up i.e. if universal educational services is missing, why was it still missing after follow-up? 7) Tables 1-4-What is Connexions (not explained on first occurrence in the text)? On what basis was the content of table 1 selected- no mention of well-baby clinics, HV postpartum or routine midwifery antenatal/postnatal contacts etc. which are "specialist services with eligibility criteria but not necessarily designed for teenage mothers". It would be helpful if table 1 domains were mapped to table 3 by adding an additional column. Why were HV visits dichotomised at 3 (i.e. is there a statistical or clinical rationale for this)? Were contacts with social services collected for trial participants? 8) Results- 'spreadsheets circulated to multiple stakeholders were more comprehensively completed' – what were the multiple stakeholders i.e. how does this relate to process described in methods section. As well as the range, what was the average number of services per site? 9) Discussion- this could make more of the methods developed and potential for use in other studies. Could also discuss the differences between site service provision and for future research to consider use of multilevel modelling analysis. 10) General comments- P14 line 30- were HV caseloads in the areas under study larger because there was a FNP scheme in operation? Final sentence ?in wrong place (it would be good to finish with an overarching conclusion); and 3rd strengths and limitations statement- not clear how the current study substantiates this comment. Abstract could better balance the two elements of the paper i.e. not just focus on the mapping but also trial participant data e.g. in the design section, including this sample size etc. I hope that these comments are helpful in supporting the researchers' dissemination of their work.
--	---

REVIEWER	Mary Barker, Associate Professor of Psychology MRC Lifecourse Epidemiology Unit, University of Southampton, UK
REVIEW RETURNED	14-Nov-2017

GENERAL COMMENTS	This is an excellent, original and thoughtful paper. The authors are to be commended for their attempt to describe usual care in terms of services for young pregnant women and for examining the functionality of this type of usual care as the control condition in a complex intervention. I enjoyed reading this paper. Thank you for sending it to me. My only major comment is that it might be helpful, if possible, for authors to suggest ways in which future process evaluations might model the impact of the trial and of innovations in service on provision of care. They say that we should do this but it is not clear how. Other than this, I only have a few minor suggestions to make.  1. Abstract - sentence 3 of the Results section and sentence 2 of the conclusions are a bit opaque. Whilst I appreciate the need to keep the word count down, is there any way of making these a bit more easily intelligible? 2. Strengths and limitations – bullet point, could the authors add a couple of words to expand on the type of bias they are talking about? 3. Methods – Eliciting and mapping services – a couple of small typos in this paragraph. 4. Enumerating service by trial participants – second paragraph, it might be helpful to add that the maternal characteristics are taken from data collected as part of the main trial. 5. Table 3 – ‘At least one burden’ what does this mean? 6. Table 5 – is this percentages rather than proportions? 7. Discussion – para 2, for whom is the information provision time consuming? 8. Discussion – para 4, first two sentences assume a lot of knowledge about FNP in the US. Could some more detail be added to these two sentences to help readers who don’t know so much? 9. Discussion – para 4, phrase in brackets could be deleted. Not particularly helpful for those who don’t know the US context and not necessary for the sense of the para.
---

VERSION 1 – AUTHOR RESPONSE

Response to editor and reviewers’ comments

Manuscript ID: bmjopen-2017-020152

Dear Editor

We thank both reviewers for their positive and constructive comments on our manuscript. We have amended the manuscript accordingly to address each of their points and also those raised directly by the editorial team. We have listed each point raised and our response below. Changes to the manuscript are indicated using tracked changes.

Yours sincerely
Mike Robling

Editorial Requirements:

- Please revise your title to state the research question, study design, and setting (location). This is the preferred format for the journal.

- We have modified the title to match the journal requirements

- Please revise the Strengths and Limitations section (after the abstract) to focus on the methodological strengths and limitations of your study rather than summarizing the results.

- We have modified this section to focus upon methodological strengths and weaknesses

- Please complete and include a STROBE check-list, ensuring that all points are included and state the page numbers where each item can be found: the check-list can be downloaded from here: <http://www.strobe-statement.org/?id=available-checklists>

- We have now provided a completed STROBE check-list (attached)

Reviewer(s)' Comments to Author:

Reviewer: 1

Reviewer Name: Janice Christie

Institution and Country: The University of Manchester, Oxford Road, Manchester, M13 9PL

Please state any competing interests: None

Please leave your comments for the authors below

Thank you for the opportunity to review this paper. It addresses an important gap in the literature about usual postpartum care is for UK postpartum teenage and young first-time mothers. In addition, it also provides a potential method of collecting information for any trial in which there is a complex context and/or potentially wide variation in usual care.

It would be helpful to clarify or add additional information about:

1) The context of postpartum care especially for teenage and young first-time mothers in the UK for the international reader. This could include a brief description/evaluation of the domains in table 1.

- We have added an extended footnote to Table 1 describing the structured of health (including health visiting), education and social services provision in England for the benefit of international readers.

2) Clearer conceptualisation in the paper about what usual care is and the difference between what is available versus what is used. E.g. Line 26 refers to universal health services; yet, the paper considers a wider range of services that could impact health. The mapping tool identified what was available; yet, the trial data identified what was used. It would be helpful to add a column to table 3 with the table 1/mapping tool main categories.

- We agree this is an important distinction. As services actually used are reported across multiple tables, we have therefore elaborated on this point in the Discussion, including that under/non-accessed but available services also represents potential for future benefit.

3) Methods/sample- how many sites were mapped (this is in the abstract and it would be helpful to have a reminder around lines 35-38). Some contextual information about the sites would be great e.g. rural/urban, geographically close or dispersed (with the same Trusts) and socio-economic status. What was the age range of the trial sample and were the 'usual' services available to everyone in this age range (it appears some universal services are missing from table 1 e.g. well baby clinics and antenatal/postpartum home visits by HV and midwives)? It would be helpful to clarify if data was

sought about universal services that only supported younger parents or if it also included services that were available to every parent.

- We have provided further information about the sites in the first paragraph of the Methods section.
- We have added details of trial eligibility criteria in the Methods section and ages of the trial sample in Results.
- The mapping exercise sought information on all services that would be available to women within this age range (which would include some services that would be available to women of all ages). Table 1 provides examples of services described and in doing so we have focused on specialist or locally available services for younger or teenage mothers.
- We have added clarification to the Methods section that services could include those also available (eg universally) to older women.

4) Methods/data collection- did the mapping tool require the site contacts to provide a description and title of service? How many mapping tools were initially disseminated per site and what was the role and background of the site contacts e.g. were they joint research and clinician posts? If they had a clinical role what was their role e.g. service providing health care professional or line manager or senior manager? Was data collected about how many consultations (and with whom) site contacts made to complete the form? If so was this similar or did it vary across sites? When was mapping data collected? When was the service use data collected from participants?

- The details of what was requested in the mapping tool are provided in Appendix 1.
- We have clarified in Methods that the mapping tool was initially sent to the local site principal investigator who cascaded further to local service leads / managers for completion, and have clarified the roles of those completing it.
- We describe in Results that at least six informants per site contributed to service elicitation in round 1 (ie completing the form). In some cases more than one person may have helped provide information to the one person collecting information for a domain but that would not have been routinely recorded.
- We have added relevant dates for service elicitation (both rounds 1 and 2) and participant service usage to the body text of the manuscript (this was already provided in table 1).

5) Methods/data analysis- Could 'summative content analysis' be more clearly explained? For example, does this refer to content analysis e.g. counting the occurrence of words/phrases or development of themes to group together similar types of services?

- The second approach listed by the reviewer reflects the approach we took in practice. We agree that our original description was insufficiently clear and therefore we have simplified and have provided some further detail in the analysis plan.

6) Methods/missing data-What were the criteria for follow up of missing information? For example was the total number of services counted and areas that looked like they had few services contacted for follow up or was this process more thematic? How much data was originally missing (e.g. how much was picked up on the 2nd round)? Was the 2nd round of mapping sent to the same people as before and did people respond to follow up i.e. if universal educational services is missing, why was it still missing after follow-up?

- No hard criteria were used to follow up missing data, but for example may have occurred if other information from the same site (eg documentary evidence or from another domain) suggested that there may be a gap.
- We have not otherwise quantified level of missing data. Apparent missing data may also not have been missing, rather that there was no (separate) service actually in existence.
- The second round of the service mapping was to FNP supervisors. They were different to those participants in round one but more likely to have a broad knowledge of applicable services locally available.

- We chose not to specifically follow up for core universal services (such as mainstream education) and instead on services that had a focus on the study population (eg FE college with special provision for teenage mothers).

7) Tables 1-4-What is Connexions (not explained on first occurrence in the text)? On what basis was the content of table 1 selected- no mention of well-baby clinics, HV postpartum or routine midwifery antenatal/postnatal contacts etc. which are “specialist services with eligibility criteria but not necessarily designed for teenage mothers”. It would be helpful if table 1 domains were mapped to table 3 by adding an additional column. Why were HV visits dichotomised at 3 (i.e. is there a statistical or clinical rationale for this)? Were contacts with social services collected for trial participants?

- We have now provided a description of Connexions when first used in the text.
- ‘Locally available services ...’ would exclude universally available services which may be provided across all sites (whether provided specifically for women of a certain age or all women). Hence, such routine midwifery care (for example) would not be reported here.
- We have added a footnote to table 1 to clarify this. We are not sure how domains would be mapped to table 3 (which provides results of the analysis of predictors of health visitor visits). Could this be further clarified?
- The clinical rationale for dichotomising the distribution of HV visits is to distinguish between standard and enhanced care. We have added some description for this.
- Contacts with social services are included under Other services in table 5.

8) Results- ‘spreadsheets circulated to multiple stakeholders were more comprehensively completed’ – what were the multiple stakeholders i.e. how does this relate to process described in methods section. As well as the range, what was the average number of services per site?

- Within a domain either one person or more than person could have contributed to its completion. However, only one person was responsible for its final completion per site / domain. We have added that detail to the Methods section.
- We have not included average number of services per site as we do not think that adds clarity when individual services can vary in both scope and scale.

9) Discussion- this could make more of the methods developed and potential for use in other studies. Could also discuss the differences between site service provision and for future research to consider use of multilevel modelling analysis.

- We have added some further commentary on the method and use in other studies in Discussion.
- We have added a comment in Discussion about the facilitation of approaches such as multi-level modelling through detailed service mapping

10) General comments- P14 line 30- were HV caseloads in the areas under study larger because there was a FNP scheme in operation? Final sentence ?in wrong place (it would be good to finish with an overarching conclusion); and 3rd strengths and limitations statement- not clear how the current study substantiates this comment. Abstract could better balance the two elements of the paper i.e. not just focus on the mapping but also trial participant data e.g. in the design section, including this sample size etc.

I hope that these comments are helpful in supporting the researchers’ dissemination of their work.

- HV caseload would not likely be importantly greater with the presence of FNP at that site given the small maximum number of concurrent FNP clients.
- We have edited the final paragraph of the Discussion along the lines suggested.

- We have modified this point to focus upon the need to revisit service maps to accommodate changes in services and added some further text in Discussion related to changes that were subsequently observed in the Connexions service (as an example).
- We have edited the Abstract to better reflect balance to the study's content.

Reviewer: 2

Reviewer Name: Mary Barker, Associate Professor of Psychology

Institution and Country: MRC Lifecourse Epidemiology Unit, University of Southampton, UK

Please state any competing interests: None declared

Please leave your comments for the authors below

This is an excellent, original and thoughtful paper. The authors are to be commended for their attempt to describe usual care in terms of services for young pregnant women and for examining the functionality of this type of usual care as the control condition in a complex intervention. I enjoyed reading this paper. Thank you for sending it to me.

My only major comment is that it might be helpful, if possible, for authors to suggest ways in which future process evaluations might model the impact of the trial and of innovations in service on provision of care. They say that we should do this but it is not clear how.

- We have added a paragraph to the Discussion to expand upon some relevant theoretical perspectives and Methods that address this good point.

Other than this, I only have a few minor suggestions to make.

1. Abstract - sentence 3 of the Results section and sentence 2 of the conclusions are a bit opaque. Whilst I appreciate the need to keep the word count down, is there any way of making these a bit more easily intelligible?

- We have modified these two sentences to clarify the points being made.

2. Strengths and limitations – bullet point, could the authors add a couple of words to expand on the type of bias they are talking about?

- We have modified this as suggested

3. Methods – Eliciting and mapping services – a couple of small typos in this paragraph.

- We have edited this section to address feedback from both reviewers

4. Enumerating service by trial participants – second paragraph, it might be helpful to add that the maternal characteristics are taken from data collected as part of the main trial.

- We have clarified this accordingly.

5. Table 3 – 'At least one burden' what does this mean?

- We have listed the original items in the table footnote

6. Table 5 – is this percentages rather than proportions?

- We have modified the titles to both tables 4 and 5 to clarify that percentages are being presented

7. Discussion – para 2, for whom is the information provision time consuming?

- We have clarified in the text that this is for professionals or other key informants.

8. Discussion – para 4, first two sentences assume a lot of knowledge about FNP in the US. Could some more detail be added to these two sentences to help readers who don't know so much?

- These sentences refer specifically to three US trial populations and their experience of FNP (in the US the programme is called Nurse Family Partnership) rather than the broader service implementation. We have therefore provided some further detail about these specifically.

9. Discussion – para 4, phrase in brackets could be deleted. Not particularly helpful for those who don't know the US context and not necessary for the sense of the para.

- We have modified these sentences to more clearly state how not only the specific services context but also the broader socio-ecological context can be used to understand how this specialist nurse-visiting service may produce varying outcomes between the US and English context

VERSION 2 – REVIEW

REVIEWER	Janice Christie University of Manchester England
REVIEW RETURNED	17-Jan-2018

GENERAL COMMENTS	Thank you for all your hard work in preparing this paper for publication. I wish you every future success. I have 2 suggestions 1) p7 line 17 indicate that table 1 only includes examples of services aimed at YP and teenagers. 2) I apologise for my error in suggesting that domains are added to table 3- I meant table 5. This is not something that is essential to do, it would help highlight your discussion about what is available versus what is used.
--

REVIEWER	Mary Barker, Associate Professor of Psychology MRC Lifecourse Epidemiology Unit, University of Southampton, UK
REVIEW RETURNED	29-Jan-2018

GENERAL COMMENTS	Thank you to the authors for their careful and comprehensive consideration of the reviewers' comments. I have no more comments to add on this revision.
---

VERSION 2 – AUTHOR RESPONSE

Dear Editor

We thank the reviewers for assessing our revised submission and for their positive comments. We have made the following changes in response.

Reviewer 1

Suggestion 1 - clarify in text the focus of table 1 upon only young parents and teenagers.

Response: We have amended the text description accordingly.

Suggestion 2 - add service mapping domains identified in table 1 to table 5.

Response: The reviewer did not consider this essential and we have not made this change. A range of services were identified by professional key informants across sites but not all services would necessarily have been available at each site to all women (and we had not attempted to enumerate that). Therefore while low levels of service uptake were generally reported and we highlight the potential for greater use of currently available services, we would not want to suggest that each service listed was fully accessible to all women for the full duration of the study. We do not think that modifying table 5 would necessarily strengthen the Discussion point.

Reviewer 2

No changes recommended.